# International Trends in Lithium Use for Pharmacotherapy and Clinical Correlates in Bipolar Disorder: A Scoping Review

**DOI:** 10.3390/brainsci14010102

**Published:** 2024-01-20

**Authors:** Yao Kang Shuy, Sanjana Santharan, Qian Hui Chew, Kang Sim

**Affiliations:** 1Lee Kong Chian School of Medicine, Nanyang Technological University, Singapore 308232, Singapore; shuy0004@e.ntu.edu.sg; 2Department of Emergency and Crisis Care, Institute of Mental Health, Singapore 539747, Singapore; sanjana.santharan@mohh.com.sg; 3Research Division, Institute of Mental Health, Singapore 539747, Singapore; qian_hui_chew@imh.com.sg; 4Yong Loo Lin School of Medicine, National University of Singapore, Singapore 117597, Singapore; 5West Region, Institute of Mental Health, Singapore 539747, Singapore

**Keywords:** bipolar disorder, lithium, mood stabilizer, clinical correlates

## Abstract

Lithium remains an effective option in the treatment of bipolar disorder (BD). Thus, we aim to characterize the pharmaco-epidemiological patterns of lithium use internationally over time and elucidate clinical correlates associated with BD using a scoping review, which was conducted using the methodological framework by Arksey and O’Malley (2005). We searched several databases for studies that examined the prescriptions for lithium and clinical associations in BD from inception until December 2023. This review included 55 articles from 1967 to 2023, which collected data from North America (*n* = 24, 43.6%), Europe (*n* = 20, 36.4%), and Asia (*n* = 11, 20.0%). The overall prescription rates ranged from 3.3% to 84% (33.4% before and 30.6% after the median year cutoffs). Over time, there was a decline in lithium use in North America (27.7% before 2010 to 17.1% after 2010) and Europe (36.7% before 2003 to 35.7% after 2003), and a mild increase in Asia (25.0% before 2003 to 26.2% after 2003). Lithium use was associated with specific demographic (e.g., age, male gender) and clinical factors (e.g., lower suicide risk). Overall, we found a trend of declining lithium use internationally, particularly in the West. Specific clinical correlates can support clinical decision-making for continued lithium use.

## 1. Introduction

Bipolar disorder (BD) is an affective disorder that can have an episodic, recurrent course and consists of a spectrum of mood changes between elated mood and depressive mood [1]. The prevalence of BD is about 0.6% for bipolar I disorder and 0.4% for bipolar II disorder [2]. Bipolar I disorder has a similar prevalence for males and females, while bipolar II disorder is observed more commonly in females [3]. As BD is potentially a long-term and recurrent illness, it is important to focus both on acute management and long-term maintenance of treatment to minimize illness relapse [4]. Of note, the suicide rate among patients suffering from BD is found to be about 20–30-fold greater than the general population [5], and comorbid psychiatric and medical conditions are higher in patients suffering from BD with an overall high illness burden [3]. A cross-sectional study conducted in the United States revealed that patients suffering from BD, especially those with moderate to severe bipolar depression, had increased hospitalization and higher direct as well as indirect costs of treatment [6]. This highlights the high burden of disease and the pressing need to evaluate the treatment options available in the management of BD.

There are various drugs available for the treatment of BD [7,8], with lithium being one of those with the longest history of use in clinical settings. Lithium is a mood stabilizer that was first introduced in 1949 and remains the best-established long-term treatment for BD [7]. Lithium exerts an overall neuroprotective effect through the upregulation of brain-derived neurotrophic factor (BDNF) [9], which prevents cellular degeneration [9] and is associated with the severity of manic and depressive symptoms in BD patients [10,11]. Lithium has also demonstrated effectiveness in managing other symptoms related to BD, including the reduction in suicidal tendencies among patients suffering from this illness [12,13].

However, there were previous studies that observed that the rate of prescription for lithium had decreased over the years [14,15], despite being prescribed for a longer duration compared to other pharmacotherapeutic agents [16]. This trend of reduced lithium prescriptions has been observed internationally. A study in Germany found that the prescriptions of lithium fell from 31.4% to 26.2% from 2009 to 2018 [17]. Even in countries like the United States, where lithium is less commonly used [18,19], lithium prescription rates have experienced a steady decline. At the Stanford University Bipolar Disorder Clinic, lithium use in BD saw a drop from 27.6% to 22.7% between 2000 and 2011, while the prescription rates of alternative medications such as lamotrigine, quetiapine, and aripiprazole doubled across the same period [20].

Some reasons for this trend of decreased lithium use include its adverse effects and narrow therapeutic index [21], with a significant risk of toxicity at levels only 2–3 times above its therapeutic dose [22]. These concerns have been validated by physician opinion studies, which found low adherence to serum-level monitoring and side effects to be the most significant considerations that influenced lithium prescription in their practice [23,24]. There are also possible adverse effects on renal function and fetal growth, and its use in populations such as the elderly and pregnant women requires careful consideration of the risks and benefits [22]. Lithium was found to have a teratogenic effect, and its use was associated with an increased risk of cardiac malformations (characteristically the Ebstein anomaly) [25,26], preterm births [27], as well as postnatal effects such as gastrointestinal bleeding and hypothyroidism [28,29].

In view of the efficacy of lithium in treatment of BD and clinical impressions of decreased reported prescription trends, we sought to conduct a scoping review of extant international studies involving lithium use in BD, including its prescription rates cross-sectionally and/or longitudinally and relevant clinical associations.

## 2. Materials and Methods

This scoping review was conducted using the 5-step methodological framework by Arksey and O’Malley (2005), namely identifying the main research questions of interest, identifying relevant studies, conducting a study selection, charting the data, and finally collating, summarizing, and reporting the results. The first step involved identifying the main research questions of interest, which were as follows: (a) What was the prescription rate of lithium in BD, cross-sectionally and longitudinally if available, and (b) What were the relevant clinical correlates of lithium prescription (such as socio-demographic variables, treatment setting, illness features, and biological factors)? The second step involved identifying relevant studies for inclusion in the review. We searched several databases (including PubMed, Medline, and Scopus) for relevant studies that examined the prescription rate of lithium or clinical correlates within BD from database inception until October 2023. The third step involved study selection. The search strategy used for this review can be found in Appendix A.

Articles were included if they were (a) cross-sectional or cohort studies that reported (b) lithium prescription rates or clinical correlates associated with lithium use within (c) patients formally diagnosed with BD. Articles were excluded if they (a) were not published in peer-reviewed journals, (b) were published in any language other than English, or (c) were conference proceedings, protocol papers, or preprints. Lithium prescription rates were included if they represented the percentage of overall BD patients that received lithium, regardless of whether it was within the context of monotherapy or combination therapy. The articles were screened by two independent reviewers using Covidence (Melbourne, Australia).

We then proceeded with the fourth step, which involved charting the data. The reviewers sequentially evaluated the titles, abstracts, and then full texts of all the publications identified by the searches for potentially relevant articles. Information regarding publication details (author, year of study, country), study details (including study population information such as age, gender, treatment setting), and main findings (prescription rates of lithium, clinical correlates associated with lithium use) were extracted. Finally, the fifth step involved collating, summarizing, and reporting the results, and the main features of all included articles are summarized and presented in Table 1.

Subgroup analysis was conducted to look for trends in lithium use across 3 main world regions (North America, Europe, and Asia), which were associated with all the included studies. The average prescription rates for each world region were calculated and weighted based on the sample size of each study. For subgroup analysis of each world region, we analyzed the lithium prescription rate before and after a predetermined cutoff year based on the median year of the study samples included across time to ensure comparable sample subjects were examined. The overall average prescription rates for the 3 world regions were also calculated. This analysis was performed to account for the differences in sample size and year(s) of sampling to better represent the differences across subgroups. For studies that reported the trends of lithium prescriptions over time with 3 or more valid data points, we evaluated the statistical significance of these trends using a linear regression model. The correlation coefficient was analyzed with 95% confidence intervals as continuous measures. All statistical analyses were performed using the Statistical Package for the Social Sciences (SPSS 22; IBM Corporation, Armonk, NY, USA).

## 3. Results

Overall, a total of 797 records were obtained from the searches. After screening, a total of 99 reports were assessed for eligibility (Figure 1). Of these, 47 were excluded, and the remaining 55 articles were included in this scoping review.

Table 1 summarizes the results from all 55 studies that were included in this scoping review, of which 45 included a valid prescription rate between 1967 and 2020.

A total of 317,851 patients were sampled within the 55 included studies. Data was collected from North America (*n* = 24, 43.6%), Europe (*n* = 20, 36.4%), and Asia (*n* = 11, 20.0%). The mean age of the participants sampled was 46.1 years old. Female patients made up 46.8% of the overall study sample. Inpatient settings were sampled in 29 (64.4%) studies, while outpatient settings were sampled in 26 (57.8%) studies. Most of the studies were published after 2010 (*n* = 38, 69.1%), followed by 2000–2009 (*n* = 11, 20.0%), 1990–1999 (*n* = 5, 9.1%), and before 1989 (*n* = 1, 1.8%). The overall weighted average lithium prescription rate across all the studies was 32.0%.

### 3.1. By Region

In view of the wide distribution of international samples across studies over time, we first grouped the included extant studies under 3 world regions, namely North America, Europe, and Asia. Next, within each world region, we took the median year of the study samples included across time to ensure comparable sample subjects were examined and reported the rates of lithium use before and after the median year cutoff (Table 2). The median year cutoff for North America was 2010, while the median year cutoff for Europe and Asia was 2003. Across the world, there was an observed decline in lithium prescription rates from 33.4% before the cutoff year to 30.6% after the cutoff year. This decline was the most substantial in North America, whereby the lithium prescription rate declined from 27.7% pre-2010s to 17.1% post-2010s. Lithium use in Europe was the highest out of the 3 world regions, both before (36.7%) and after (35.7%) the cutoff year. There was a mild increase in overall lithium prescription rates in Asia (from 25.0% before 2003 to 26.2% after 2003).

### 3.2. By Time

Sixteen studies reported multiple data points regarding lithium prescription rates over time, of which 11 provided sufficient information for linear regression analysis. All studies reported a decreasing trend in lithium prescription rates, except for Kleimann et al. (2016) [51], which reported a non-significant increase in prescription rates. This declining trend is statistically significant in 80% of the remaining studies [14,15,35,38,42,49,52,61].

### 3.3. By Special Demographics

A total of 7 studies were found reporting prescription patterns of lithium use in bipolar patients with specific demographic profiles. Five studies investigated children or adolescents, while the other 2 studies included pregnant women and the elderly, respectively. The weighted average lithium prescription rate was 16.0%, 18.5%, and 70% for children and adolescents, pregnant women, and the elderly, respectively.

### 3.4. By Clinical Setting

We calculated the weighted lithium prescription rate between inpatient and outpatient treatment settings. A total of 12 studies (21.8%) reported lithium prescription rates from inpatient settings only, while another 13 studies (23.6%) sampled prescription rates from outpatient settings only. The remaining studies (*n* = 30, 54.5%) either sampled from both treatment settings or did not report the clinical setting within the study. Inpatient settings were associated with an average weighted prescription rate of 37.1%, while outpatient settings were associated with an average weighted prescription rate of 37.6%.

### 3.5. By Treatment Regimen

There were few studies that reported the rates of patients on lithium monotherapy and those on adjunctive lithium therapy. For use as adjunctive therapy, lithium was noted to be commonly prescribed in conjunction with antipsychotics (11.0–21.0%) [58,65] or anticonvulsants (6.5%) [58], with some patients being prescribed a combination of all 3 drugs (7–11%) [58,65]. Rates of lithium monotherapy fell by 41% over 5 years at a North American site [78]. Lithium was still the most prevalent drug used for monotherapy in one study, with 10.4% of BD patients being on this regimen [58]. Nonetheless, two other studies reported rates of valproate monotherapy (44.2%) surpassing that of lithium monotherapy (38.9%) [37], and rates of atypical antipsychotic monotherapy (62–71%) surpassing that of lithium monotherapy (5–12%) [65]. Patients on lithium or valproate monotherapy had a significantly lower likelihood of all-cause hospitalizations compared to patients on second-generation antipsychotics [47].

### 3.6. Clinical Correlates of Lithium Prescription

Several studies investigated factors that were associated with BD patients taking lithium. Lithium use was found to be associated with demographic factors such as male gender [14,42,55,62] and non-blacks [71]. The relationship between age and lithium use was less consistent, in that 2 studies found that lithium users were associated with older age [14,42], but 2 other studies reported associations with younger age [51,68]. Another study among adolescents with BD in an outpatient setting found that older adolescents were more likely to be prescribed lithium [57]. Lithium use was also associated with several clinical characteristics of patients, including psychotic symptoms [57] and lower suicidal risk [54,57,70]. Of note, lithium users had a lower likelihood of hospitalization [30,47,57], a lower risk of cancer [48] and stroke [53], as well as lower levels of HbA1c [36] and triglycerides [36] compared to patients not on lithium. A summary of some of the main findings from this review can be seen in Figure 2.

## 4. Discussion

There were several findings from our review. First, the overall average lithium prescription rate across all studies was 32.0%, being highest in Europe, followed by North America and Asia. Second, there was a noted decline in lithium prescription rates over time, especially in North America and Europe. Third, there was no difference between lithium prescription rates in inpatient compared to outpatient settings, and lithium use was associated with certain demographic and clinical factors.

There are notable differences in the prescription rate of lithium across the three world regions, both cross-sectionally and over time. Europe had the highest lithium prescription rate of above 35% both before and after the cutoff year. Before the cutoff, North America (27.7%) and Asia (25.0%) had comparable prescription rates. However, after the cutoff year, the Americas (17.1%) had a significantly lower lithium prescription rate compared to Asia (26.2%) and Europe (35.7%). The reasons underlying these international differences are likely multifaceted and complex, such as the nature of patient profiles, illness onset and course, healthcare system considerations, including the availability of newer drugs, and patient and practitioner preferences. For example, a retrospective study comparing the clinical characteristics of BD-1 patients in the United States compared to France found that American patients were more likely to present with a depressive onset compared to French patients (71% vs. 57.9%) [81]. Given that lithium is not recommended as a first-line treatment for patients with acute bipolar depression based on several international treatment guidelines for BD [82], this could result in a lower lithium prescription rate in North America. In the same study, a greater proportion of American patients were also classified as early-onset compared to French patients (68% vs. 42%) [81]. From the included samples within this scoping review, it was noted that the prescription rate of lithium in early-onset BD was low, between 5 and 10% [63,66], which may account for the lower lithium prescription in the Americas versus Europe.

Over time, the overall prescription rate of lithium changed from 33.4% to 30.6% across the cutoff period, suggesting a small decline in the use of lithium over time across our varied samples. This shift is more prominent when comparing trends in lithium use over time within individual studies. Several studies that reported lithium prescription rates individually across multiple time points showed a significant decrease in lithium prescription rate over time [14,15,35,38,42,49,52,61]. The fall in lithium prescription rates can be viewed in the context of a concomitant rise in alternative medications such as second-generation antipsychotics (SGAs) being used as mood stabilizers. The use of SGAs more than doubled between 1996 and 2007 in the United States [38], a trend that is strongly observed across multiple studies [41,83,84]. While studies have demonstrated that lithium has similar efficacy compared to SGAs such as quetiapine in the setting of acute mania [85], lithium may be deemed clinically less favorable by clinicians and/or patients due to its narrow therapeutic window necessitating close drug monitoring [86] and potentially undesirable side effects such as tremors, weight gain, and hypothyroidism [21]. In addition, patients may be less adherent to lithium treatment due to the above-mentioned potential adverse effects and burden of drug monitoring, thus further rendering it a less popular prescription option despite its proven clinical efficacy [35]. In our review, the fall in lithium prescription rates was the greatest in the Americas (27.7% to 17.1%), with a small decrease in Europe (36.7% to 35.7%) over time. Although the prescription rate of lithium increased slightly in Asia (25.0% to 26.2%), there are likely inter-site variations as sites such as Hong Kong [35] and Taiwan [49] reported relatively more marked decreases in prescription rates from 25.8% (2003) to 17.6% (2018) and 18.3% (2001) to 6.9% (2010), respectively. The continuous decline and underuse of lithium over time has been noted in the recent literature [87,88], but lithium is still the drug of choice in various treatment guidelines [89,90,91], especially for its efficacy in acute management and maintenance therapy in BD as well as its anti-suicidal effects [92,93]. Contrary to earlier concerns, there are data to suggest that BD patients on lithium may be more likely to continue taking the medication compared to other alternatives once they have experienced its beneficial effects [94,95]. As such, clinical decision-making regarding lithium use needs to consider patient concerns, previous experiences, and clinical considerations, and there is a need for adequate discussion about the indications, adverse effects, drug monitoring, close reviews, and risk–benefit analyses bearing these factors in mind [96,97,98].

Few studies clearly delineated the rates of patients on lithium monotherapy versus adjunctive lithium therapy. Those that reported on such variations in lithium use observed a high prevalence of polypharmacy use across sites, which could be attributed to the increase in treatment-resistant BD patients [31], elevated tolerance to lithium [99,100,101], greater illness chronicity [58], or limited efficacy of extant treatments for BD [31]. This is consistent with existing treatment guidelines that support the use of combination therapy for treatment-resistant BD [13,89,102], which was associated with reduced hospitalizations, decreased time to recurrence, and improved quality of life for patients with BD [103,104].

There were several clinical correlates associated with lithium use. Regarding demographic factors, lithium was associated with the male gender [42,55,62]. Males tended to have an earlier onset of illness [105,106] and more manic symptoms than females [107], both of which are factors that support the use of mood stabilizers, including lithium. The association of lithium use with age was less consistent. In a sample of 32,019 patients with BD, lithium users were on average 5.4 years older than non-users [14], which agreed with the findings of Lyall et al. (2019). In contrast, two other studies found that lithium was more commonly prescribed to younger patients [51,68]. Possible reasons for the association with older age at the time of sampling could be the chronic nature of the illness and the need for lithium maintenance treatment. The risk of nephrotoxicity [108,109,110] in older patients with compromised renal function could be a possible reason for the association with younger patients, since lithium is less often prescribed for older BD patients.

Regarding clinical factors, a study found that adolescent patients with psychotic features and self-injurious behavior were more likely to be on lithium [57]. A meta-analysis of 31 observational studies found that BD patients on lithium had a significantly lower risk of suicidal behavior [70], which was in line with the extant literature, which found that the time to self-harm was the longest for patients on lithium compared to other mood stabilizers and antipsychotics [111]. However, evidence on the use of lithium in patients with psychotic features is less consistent. While lithium has been shown to help BD patients with psychotic features [112], BD patients with psychotic features were also found to have a poorer response rate to lithium [113]. More recently, it was reported that lithium and quetiapine were both found to have similar treatment outcomes when used in acute bipolar depressive episodes with psychotic features [114]. The positive association between lithium and psychosis may be due to the overall clinical profile of BD patients with psychotic features, as these patients may have more severe episodes and higher suicidality [115], which necessitates the initiation of lithium.

In addition, patients on lithium monotherapy were found to have a significantly lower likelihood of all-cause hospitalizations and mental health hospitalizations than SGAs [47,57]. In a separate study, the early initiation of lithium in BD patients was found to have decreased the risk of hospitalization [116]. This observed relationship can be attributed to how lithium prolonged the time to intervention for manic or hypomanic episodes across multiple studies [117,118,119] and improved symptom control and illness management, thus leading to improved clinical outcomes, including fewer hospitalizations. Apart from its positive effects on clinical outcomes, lithium was associated with lower risks of cancer [48], a reduced risk of stroke [53], and insulin resistance [36]. Given that patients with BD are at increased risk of cancer [120], stroke [121,122,123], and obesity [124], the potential protective effects of lithium on medical conditions such as cancer and stroke can be taken into consideration for patients who were assessed to have a family history of such conditions after further warranted studies with replication. There were relatively fewer reports of severe weight gain in patients treated with lithium as compared to those treated with an SGA, although it was also noted that there was a relatively higher incidence of severe weight gain reported in users under 45 years old on lithium as compared to non-lithium users [125]. It has also been postulated that patients with BD and insulin resistance may denote a subgroup of patients with more severe clinical symptoms and a poorer response to lithium [126,127]. The effect that lithium has on fetal growth and development may also have been overstated in previous studies. A recent study concluded that the magnitude of cardiac malformations in infants exposed to lithium was smaller as compared to rates in the past [128], suggesting that lithium can still be used during pregnancy, albeit not as first-line therapy if clinically indicated, but needs careful monitoring and review [129,130]. Thus, greater attention to the clinical profiles of BD patients may help clinicians identify subtypes of patients that can benefit from or respond to lithium more than others.

There were some limitations to this study. First, there was wide heterogeneity in the included data across the studies and paucity in certain areas, such as BD subgroups and details of illness courses, which preclude further examination of clinical correlates. Second, there were more studies conducted in the West (North America and Europe) compared to Asia. Third, not all studies were conducted prospectively to determine the changes in lithium prescription over time. Future research is warranted to examine longitudinal trends and clinical and biological markers associated with lithium use to better optimize the pharmaco-therapeutic management of BD patients seen in our clinical settings.

There are several future directions that are worthy of consideration. First, given the suggested benefits of pharmacotherapy using lithium, such as reduced risk of hospitalization and improved symptom control, future studies should seek to follow up on such clinical outcomes in larger patient cohorts over a longer period of time to further discern clinical predictors of a positive or suboptimal response. Such studies can help to highlight subgroups of BD patients who may respond better or worse to lithium treatment (whether lithium monotherapy or combination therapies). Second, a better understanding of the biological correlates of the lithium response (such as pharmacogenetic factors, neuroanatomical substrates, neurophysiological measures, etc.) can improve the biological understanding of the response to lithium treatment. Third, the basis of observations such as reduced cancer and stroke risk needs better elucidation. Fourth, a better clinical understanding of lithium in the treatment of resistant BD can enhance clinical treatment in patients with greater illness burden. These suggested areas of research will allow us to work towards achieving the promise of precision and personalized medicine, with the potential to deliver effective treatments for patients with BD in a timely manner and minimize the process of trial and error in arriving at the most appropriate management option.

## 5. Conclusions

In conclusion, we found continual prescription of lithium in the management of patients with BD despite its narrow therapeutic window and need for therapeutic drug and hematological monitoring. However, there was a noted decline in lithium use globally, especially in the West. There were observed associations between lithium use and certain demographic and clinical factors in BD patients. Future studies are warranted to examine physician attitudes toward the prescription of psychotropic drugs for BD patients over time, which can uncover relevant reasons and concerns that may account for the clinical use of such medications, including lithium, for the treatment of patients with BD. This review also prompts further research to better delineate longitudinal trends and clinical and biological markers associated with lithium use to optimize the treatment of BD patients entrusted to our care.

## Figures and Tables

**Figure 1 brainsci-14-00102-f001:**
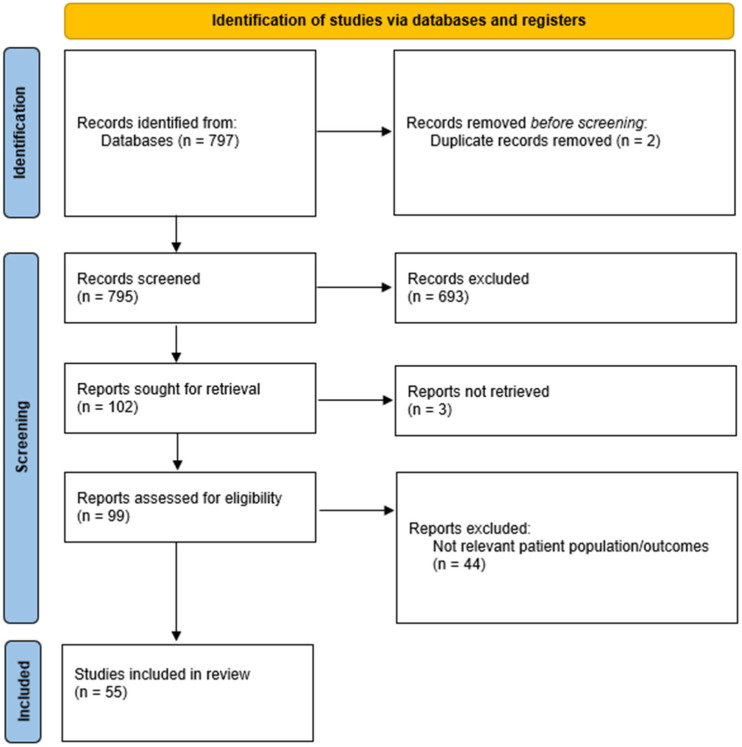
PRISMA flow diagram to show the process of identification and screening to determine the included studies.

**Figure 2 brainsci-14-00102-f002:**
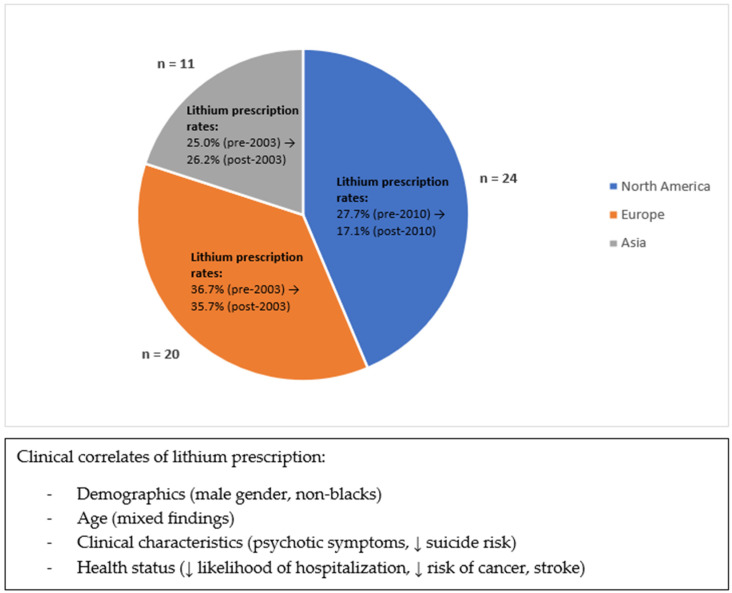
Summary of some main findings.

**Table 1 brainsci-14-00102-t001:** Summary of study details for all included papers.

Publication Details		Study Details								Main Findings	
Study	Country	Study Design	N	Mean Age (SD)	Percentage of Females	Inpatient (IP) or Outpatient (OP)	Special Demographic	Diagnosis	Year of Study Sample	Prescription Rate (PR) for Lithium (CI)	Clinical Correlates
Lähteenvuo et al., 2023 [30]	Finland	Longitudinal	60,045	41.7 (15.8)	56.4%	IP OP	NA	BD	1996–2018	NA	Lithium users were associated with a lower risk of hospitalization for psychiatric reasons than those not using medication from the same medicine class.
Singh et al., 2023 [31]	North America, Europe, and Australia	Cross-sectional	7748	41.4	60.6%	IP OP	NA	BD	1998–2020	30.3%	
Lin et al., 2023 [32]	Taiwan	Longitudinal	420	17.2 (1.9)	53.8%	OP	Children (below 20)	BD	2006–2019	23.1%	
Shinozaki et al., 2022 [33]	Japan	Cross-sectional	2563	50.7 (13.8)	54.1%	OP	NA	BD	2017	47.5% (ALL) 55.1% (BD-I) 43.2% (BD-II)	
Uwai and Nabekura, 2022 [34]	Japan	Cross-sectional	3521	Not reported	58.9%	Not reported	NA	BD	2004–2020	NA	Lithium use was not associated with Parkinson-like events.
Ng et al., 2021 [35]	Hong Kong and the UK	Cross-sectional, cohort (time trend)	HK: 15,287 UK: 30,140	HK: 38.92 (22.46) UK: 44.18 (26.69)	HK: 60.5% UK: 61.4%	IP OP	NA	BD	2001–2018	Hong Kong 2001—20.1% 2002—24.4% 2003—25.8% 2004—25.8% 2005—25.8% 2006—25.0% 2007—23.8% 2008—22.9% 2009—22.1% 2010—21.3% 2011—20.5% 2012—20.1% 2013—19.0% 2014—18.6% 2015—17.9% 2016—17.8% 2017—17.9% 2018—17.6% *** United Kingdom 2001—30.7% 2002—30.0% 2003—29.1% 2004—28.9% 2005—27.5% 2006—26.4% 2007—25.6% 2008—24.3% 2009—23.0% 2010—21.8% 2011—20.9% 2012—19.3% 2013—18.7% 2014—17.7% 2015—16.9% 2016—16.5% 2017—16.5% 2018—16.1% ***	
Prillo et al., 2021 [36]	Canada	Cross-sectional	129	Li users: 49.05 (11.78) Non-Li users: 46.71 (11.20)	Li users: 56.1% Non-Li users: 46.0%	OP	NA	BD	NA	NA	Lithium use was associated with significantly lower HbA1c and triglyceride levels. Lithium use was not associated with obesity, BMI, metabolic syndrome, hypertension, or thyroid disease.
Grover et al., 2021 [37]	India	Cross-sectional	773	45.7 (10.5)	36.4%	OP	NA	BD	Not stated	38.9% on lithium monotherapy	
Lin et al., 2020 [38]	USA	Cross-sectional, cohort (time trend)	5400	Not reported	61.6%	OP	NA	BD	1996–2015	1996–1997: 38.0% 1998–1999: 28.2% 2000–2001: 22.7% 2002–2003: 18.9% 2004–2005: 21.8% 2006–2007: 14.4% 2008–2009: 12.5% 2010–2011: 12.6% 2012–2013: 14.1% 2014–2015: 14.7% **	
Karanti et al., 2020 [39]	Sweden	Cross-sectional	8766	BD I 50.2 (15.7) BD II 45.38 (15.6)	61.9%	OP	NA	BD	2004–2013	BD I 68.6% BD II 44.7%	
Salazar de Pablo et al., 2020 [40]	USA	Cross-sectional	76	15.6 (1.4) (12–18)	59.2%	IP	Children (age 12–18)	BD	2009–2017	22.4% on admission 35.5% on discharge	
Bohlken et al., 2020 [17]	Germany	Cross-sectional, cohort (time trend)	4137	2009: 53.5 (15.0) 2018: 56.4 (15.0)	67.9%	OP	NA	BD	2009 and 2018	2009: 31.4% 2018: 26.2%	Non-blacks have a significantly higher prescription rate for lithium (34.2%) compared to blacks (26.2%).
Rhee et al., 2020 [41]	USA	Cross-sectional, cohort (time trend)	4419	Not reported	59.4–63.8%	OP	NA	BD	1997–2016	1997–2000: 30.4% 2001–2004: 20.7% 2005–2008: 17.3% 2009–2012: 13.9% 2013–2016: 17.6%	
Lyall et al., 2019 [42]	Scotland	Cross-sectional, cohort (time trend)	20,796	Not reported	Not reported	IP/OP	NA	BD	2009–2016	2009: 26.1% 2010: 25.5% 2011: 25.2% 2012: 24.1% 2013: 23.7% 2014: 23.1% 2015: 22.4% 2016: 21.9% ***	Lithium use was more common in males and older patients.
Musetti et al., 2018 [43]	Italy	Longitudinal	234	38.6 (12.7)	60.2%	OP	NA	BD	2002–2006	76.10%	
Jaracz et al., 2018 [44]	Poland	Cross-sectional	127	46.2 (13.8)	44.9%	IP	NA	BD	2015–2016	23.60%	
Broeks et al., 2017 [45]	Denmark	Longitudinal	336	29.85 (26.3–34.03)	100.0%	Not reported	Pregnant women	BD	1997–2012	18.5% None redeemed during pregnancy: 5.7% Redeemed during pregnancy: 29.8%	
Rej et al., 2017 [46]	Canada	Cross-sectional	1443	72.24 (5.63)	63.8%	IP	NA	BD	2006–2012	23.40%	
Bauer et al., 2016 [47]	USA	Longitudinal	27,727	Lithium only: 44.85 (13.50) Lithium + SGA: 43.50 (13.29)	Lithium only: 26.8% Lithium + SGA: 16.7%	OP	NA	BD	2003–2010	NA	Patients on lithium monotherapy and valproate monotherapy were associated with a significantly lower likelihood of all-cause hospitalizations compared to patients on second-generation antipsychotic monotherapy. The initiation of lithium or valproate was associated with a significantly lower likelihood of mental health hospitalizations than second-generation antipsychotics.
Kessing et al., 2016 [15]	Denmark	Longitudinal	3205	2000: 51.0 (35.7–64.5) 2008: 40.9 (30.3–55.6)	45.1–55.8%	IP OP	NA	Single manic episode or BD	2000–2011	2000: 41.1% 2001: 40.2% 2002: 39.0% 2003: 36.9% 2004: 30.8% 2005: 32.3% 2006: 35.0% 2007: 39.4% 2008: 38.0% 2009: 33.2% 2010: 30.8% 2011: 34.0% *	
Huang et al., 2016 [48]	Taiwan	Longitudinal	4729	Not reported	Not reported	IP OP	NA	BD	1998–2009	NA	Lithium use was associated with a significantly lower risk of cancer compared to anticonvulsant users. Higher cumulative and daily doses of lithium were significantly associated with lower cancer risk.
Chang et al., 2016 [49]	Taiwan	Longitudinal and cross-sectional (time trend)	2703	40.5 (18.8)	52.5%	IP OP	NA	BD	2001–2010	2001: 18.3% 2002: 13.9% 2003: 11.2% 2004: 13.9% 2005: 14.6% 2006: 13.2% 2007: 11.5% 2008: 10.2% 2009: 7.8% 2010: 6.9% **	
Arikan et al., 2016 [50]	Istanbul	Cross-sectional, cohort (time trend)	2001–2002: 78 2011–2012: 87	2001–2002: 36.4 (12.1) 18–73 2011–2012: 37.2 (14.4) 16–70	56.4–59.8%	IP	NA	BD	2001–2002 and 2011–2012	2001–2002: 49.1% 2011–2012: 36.5%	
Kleimann et al., 2016 [51]	Austria, Switzerland, and Germany	Cross-sectional, cohort (time trend)	1650	48.87 (14.91)	53.1%	Not reported	NA	Acute mania	2005–2012	2005: 27.5% 2006: 27.5% 2007: 23.6% 2008: 26.3% 2009: 29.3% 2010: 37.7% 2011: 31.3% 2012: 28.2%	Lithium use was significantly less common in patients over 70 years old.
Karanti et al., 2016 [14]	Sweden	Cross-sectional, cohort (time trend)	32,019	Male: 51.9 (15.3) Female: 49.6 (15.9)	61.0%	Not reported	NA	BD	2007–2013	Male: 2007: 71.2% 2008: 70.0% 2009: 67.8% 2010: 66.4% 2011: 65.1% 2012: 62.9% 2013: 59.3% *** Female: 2007: 64.0% 2008: 65.0% 2009: 62.5% 2010: 59.2% 2011: 57.1% 2012: 54.9% 2013: 52.8% ***	Lithium use was significantly more common in men than women. Lithium users were, on average, 5.4 years older than non-users.
Carlborg et al., 2015 [52]	Sweden	Cross-sectional, cohort	10,273	42.6	62.3%	IP OP	NA	BD	2006–2010	2006—40.8% 2007—40.3% 2008—37.5% 2009—35.3% 2010—32.6% **	
Lan et al., 2015 [53]	Taiwan	Cross-sectional	3681	Not reported	Not reported	IP OP	NA	BD	2001–2011	NA	Lithium use was associated with a significantly lower risk of stroke than non-users.
Toffol et al., 2015 [54]	Finland	Longitudinal	826	Not reported	Unable to access tables	IP	NA	BD	1996–2003	37%	Lithium use was associated with a lower risk of suicidal behavior and overall mortality.
Karanti et al., 2015 [55]	Sweden	Cross-sectional	7354	47.9 (23.3)	61.0%	OP	NA	BD	2004–2011	NA	Lithium use was significantly more common in men than women.
Baek et al., 2014 [56]	Korea	Longitudinal	1447	B1 MI 37.32 (1.8) B1 DI 27.59 (13.7) B1 O 37.90 (14.6) B2 DI 16.89 (16.0) B2 O 35.4 (14.3)	59.7%	IP OP	NA	BD	2009	B1 MI 37.9% B1 DI 29.6% B1 O 38.3% B2 DI 29.1% B2 O 27.5%	
Ko et al., 2014 [57]	Canada	Cross-sectional	100	Li users: 17.05 (1.10) Non-Li users: 15.98 (1.47)	Li users: 75% Non-Li: users 65%	OP	Children (age 13–19)	BD	NA	NA	Lithium use was associated with older patients, a diagnosis of BD-I, psychosis, self-injurious behavior, previous psychiatric hospitalizations, and lifetime use of antimanics/anticonvulsants/second-generation antipsychotics. Lithium use was not associated with suicide ideation or suicide attempts.
Grande et al., 2013 [58]	Spain	Cross-sectional	739	46.1 (13.7)	59.3%	OP	NA	BD	2006–2007	10.40%	Lithium monotherapy was associated with a family history of psychiatric disorders, longer duration from previous BD episodes, and younger patients.
Trivedi et al., 2013 [59]	India	Cross-sectional	100	31.0 (11.9)	18.0%	OP	NA	BD	Not stated	57.0%	
Haeberle et al., 2012 [60]	Germany, Switzerland, and Austria	Cross-sectional, cohort (time trend)	2246	Not reported	Not reported	IP	NA	Bipolar depression	1994–2009	1994–1997: 44.7% 1998–2001: 36.4% 2002–2005: 30.5% 2006–2009: 34.7%	
Greil et al., 2012 [61]	Germany, Switzerland, and Austria	Cross-sectional, cohort (time trend)	2231	57	62.0%	IP	NA	Bipolar depression	1994–2009	1994: 47.9% 1995: 54.6% 1996: 47.3% 1997: 37.8% 1998: 39.2% 1999: 44.1% 2000: 30.3% 2001: 31.8% 2002: 28.5% 2003: 30.3% 2004: 30.7% 2005: 30.8% 2006: 30.6% 2007: 37.0% 2008: 35.1% 2009: 35.7% **	
Walpoth-Niederwanger et al., 2012 [62]	Austria	Cross-sectional, cohort (time trend)	531	48.9	68.1%	IP	NA	BD	1999–2007	1999–2003: 19.3% 2004–2007: 13.3%	Lithium use was significantly more common in men compared to women.
Dusetzina et al., 2011 [63]	USA	Longitudinal	412	Not reported	53.0%	IP OP	Children (age 6–17)	BD-I	2005–2007	11%	
Post et al., 2011 [64]	USA, the Netherlands, and Germany	Cross-sectional (?)	525	US: 18.6 Europe: 24.80	Not reported	OP	NA	BD	???	US: 53.33% Europe: 81.40%	
Larsen et al., 2009 [65]	Finland, Norway, Denmark, France, Italy, Germany, Greece, the Netherlands, the UK and Ireland, Belgium, Spain, Portugal, and Switzerland	Longitudinal	3459	Nordic: 46.9 (14.00) European: 44.6 (13.39)	Not reported	IP OP	NA	BD	2002–2004	Nordic—34% Lithium monotherapy 12% AP + Lithium 11% AP + Lithium + Anticonvulsants 11% European—33% Lithium monotherapy 5% AP + Lithium 21% AP + Lithium + Anticonvulsants 7%	
Jerrell et al., 2008 [66]	USA	Longitudinal	82	6–17	52.0%	IP	Children (age 6–17)	BD-I	2003–2004	4.90%	
Baldessarini et al. et al., 2008 [67]	USA	Longitudinal	7406	35.4 (12.4)	56.5%	IP OP	NA	BD	2001–2004	9.5% at initial treatment/baseline 14.5% at final treatment/12 months	Patients given lithium as the single mood stabilizer were much less likely to receive adjunctive psychotropic agents during the following year than those whose single mood stabilizer was an anticonvulsant.
Al Jurdi et al., 2008 [68]	USA	Longitudinal	2442	Not reported	56.6%	IP OP	NA	BD	1998–2005	Total: 37.2% 20–59: 37.8% 60 and above: 29.5%	Lithium use was more common in younger patients (age 20–59) compared to older patients (over 60). Lithium dosing was higher in younger patients compared to older patients.
Baldessarini et al., 2007 [16]	USA	Longitudinal	7760	40.1	39.9%	IP OP	NA	BD	2002–2003	7.50%	
Wolfsperger et al., 2007 [69]	Germany, Switzerland, and Austria	Cross-sectional, cohort (time trend)	998	46.9	50.5%	IP	NA	BD	1994–2004	Bipolar mania: 1994–1999: 46.7% 2000–2004: 36.7%	
Baldessarini et al., 2006 [70]	Many	Meta-analysis	NA	Not reported	Not reported	Not reported	NA	BD	1970–2006	NA	Lithium use was associated with a significantly lower suicidal risk than non-users.
Kilbourne et al., 2006 [71]	USA	Cross-sectional	2958	52 (12)	10.6%	IP OP	NA	BD	2001	33.20%	
Sajatovic et al., 2004 [72]	USA	Cross-sectional	65,556	52.7 (12.7)	11.4%	IP OP	NA	BD	2001	25%	
Bhangoo et al., 2003 [73]	USA	Cross-sectional	111	10.98 (2.64)	35.0%	OP	Children (age 6–17)	BD	Unknown	51%	
Levine et al., 2000 [74]	USA	Cross-sectional	457	40 (10)	67.0%	OP	NA	BD-I	1995–1996	50.10%	
Unutzer et al., 1998 [75]	USA	Cross-sectional	1236	43.1 (14)	66.1%	IP OP	NA	BD	1995–1996	60.30%	
Sajatovic et al., 1997 [76]	USA	Cross-sectional	96	50 (12.3) (23–83)	8.3%	IP	NA	BD	1993–1995	62.50%	No difference in the length of stay between patients on lithium monotherapy compared to anticonvulsant monotherapy was observed. Use of psychotropic medication was not associated with the drug regime.
Chou et al., 1996 [77]	USA	Cross-sectional	528	42.0 (13.0)	49.1%	IP	NA	BD	1990	61%	
Fenn et al., 1996 [78]	USA	Cross-sectional, cohort (time trend)	829	Not reported	Not reported	IP	NA	BD	1989–1994	Lithium monotherapy: 1989: 84% 1994: 43%	
Sajatovic et al., 1996 [79]	USA	Longitudinal	23	Not reported	Not reported	IP	Elderly (age above 65)	BD	1992–1994	70%	
Hes et al., 1976 [80]	Israel	Cross-sectional	314	Not reported	Not reported	NS	NA	BD	1967–1974	76%	

For studies that reported trends in lithium prescription rates over time with 3 or more data points, the level of significance of the trend was calculated by means of a linear regression model. * *p* < 0.05; ** *p* < 0.01; *** *p* < 0.001.

**Table 2 brainsci-14-00102-t002:** Global prescription rates of lithium over time.

Region	Lithium Prescription Rate ^2^		Number of Studies		Sample Size (n)	
	Pre-Cutoff	Post-Cutoff	Pre-Cutoff	Post-Cutoff	Pre-Cutoff	Post-Cutoff
North America ^1^	27.7%	17.1%	13	8	19,356	19,782
Europe	36.7%	35.7%	18	13	63,581	62,592
Asia	25.0%	26.2%	6	5	12,270	10,917
Overall	33.4%	30.6%	37	26	95,207	93,291

^1^ Data from Sajatovic et al. (2004) [72] was excluded from the analysis of lithium prescription rates for North America as it has an outlier sample size of 65,556. ^2^ North America used a cutoff year of 2010, while Europe and Asia used a cutoff year of 2003.

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
