# Peer review of "International Trends in Lithium Use for Pharmacotherapy and Clinical Correlates in Bipolar Disorder: A Scoping Review"

_brainsci, 2024, doi:10.3390/brainsci14010102_

Round 1

Reviewer 1 Report

Comments and Suggestions for Authors

The current review has an interesting focus (evolution of lithium prescription rates over time), but is hampered by three critical points: 1) Unclear focus in their research questions, especially the secondary RQ’s; 2) non-transparent search strategy 3) The absence of a statistical approach significantly diminishes the paper's quality, transforming it into a monotonous compilation of prevalence rates devoid of any engaging analysis or compelling presentation of the findings.

Introduction

·      Overall, the introduction is a bit long for my taste and doesn’t get to the point quickly.

·      The use of ‘subtreshold BD’ and ‘bipolar spectrum disorder’ may be confusing. Either explain the used terminology or delete.

·      Suicide rates associated to bipolar disorder seem largely underestimated in this introduction. It is stated that it is 20-30% higher than the general population. However, to my knowledge, 10-15% of bipolar patients die by suicide, whereas this number is much lower in the general population (WHO: 4-20 per 100.000 inhabitants)

·      The introduction already cites some studies and arguments aimed towards the questions it is trying to answer in this review. The buildup to the research question(s) could be cleaner. 

·      The focus of the paper should be stated more clearly at the end of the introduction

Materials and methods

·      The focus of the ‘scoping review’ is rather vague. What do the authors mean with their secondary RQ ‘relevant clinical correlates’?

·      Why ‘scoping’ and not a systematic review?

·      Importantly, what was the search strategy? What was the search string? Please add.

·      The focus of the paper begs for a metaregression approach to investigate whether prescription rates in cross-sectional studies tend to decrease depending on publication year

Results

·      Given the broad scope of this paper, I am interested in the search strategy that only yielded 797 hits.

·      Table 1: how can data be cross sectional and be collected from 1998 to 2020? (Singh), or 2004-2020 (Uwai)

·      Table 1: this is a very non-useful way of presenting the data. Again, I think (meta)regression is the way to go. At least a more visual way should be implemented to present the data more approachable

·      It seems that ‘relevant clinical correlates’ are demographic correlates? (age, gender,…). What about Duration of Illness, symptom severity, age of onset,…?

·      The focus of 3.3 seems a bit random, as it lists a series of prevalence rates that are not always covers by the title. Whether lithium was in monotherapy or adjunctive therapy is not related to the 'clinical setting' I presume? Or demographic variables for that matter? 

Discussion

·      Overall, the discussion seems to be repetitive of the results section. It tends to stay at the surface, and is victim of the nature of the results section, which lacks any statistical approach of the data, en just sums up some prevalence rates.

·      The authors comment on which fall in prescription rate per continent is most significant. They did not analyze these results, so no comments can be made on the significance of a fall, and certainly not on the significance of intercontinental differences

Conclusion

·      Authors should be careful to comment on the decline, in line with my previous comments

Comments on the Quality of English Language

Quality of English language use is ok

Reviewer 2 Report

Comments and Suggestions for Authors

Dear Authors,

I have read with great interest your manuscript entitled "International trends in lithium use for pharmacotherapy and clinical correlates in bipolar disorder: a scoping review" since the use of lithium in the treatment of BD remains controversial.

The authors have designed and performed a scoping review to assess the prescription rate of lithium based on studies conducted in the Americas, Europe, and Asia. Although there are several limitations because of the limited availability of data, the authors concluded that there is a significant decreasing trend regarding the prescription of lithium in BD patients. 

This is not so surprising given the fact that there are many drawbacks related to lithium's safety profile. But several arguments can be listed to support its use, one of the most controversial being its effect on weight. In this review, the authors claim that lithium has some protective effect on obesity, however, they should include the recent study published by Greil et al. (10.1186/s40345-023-00313-8 ). More pharmacovigilance and real-world data should be included to support the safety profile of lithium.

Overall, the study provides a thorough analysis of the scientific evidence and proposes some interesting directions for future clinical studies. In conclusion, I recommend the publication of this manuscript after a minor revision. 

  Comments on the Quality of English Language

Minor editing of English language required.

Round 2

Reviewer 1 Report

Comments and Suggestions for Authors

The two main concerns I have remain problematic:

1) the search strategy does not seem to be sufficient. The fact that they need 'epidemiology/epidemiological' to be an explicit term in the abstract or title almost guarantees that they miss relevant papers. 'Drug Utilization/ or Drug Prescriptions/ or prescription pattern*' may also be too narrow. studies investigating this phenomenon using terms such as  "nationwide drug use", "drug sales", "prescription behavior", "national databases' could be missed. In my opinion the use of 'scoping review' instead of 'systematic review' can be seen as a way to cover for incomplete literature search, but that is just an easy way out.

2) To evaluate the evolution of prescription rates, an analysis of the data is needed, as strongly suggested. And is easily done. Nonetheless, the authors indicate they chose not to do this. Now, the prescription rates are enlisted in a table, and trends are 'seen' and discussed as if these trends are truly present, but without a proper analysis, this is pure speculation. Importantly, some of these included studies report on prescription rates but have included patients over a time span of decades, which may hamper interpretability

Because of these reasons, I would propose to reject the manuscript.

Comments on the Quality of English Language

English is fine
